# Effects of PM_10_ Airborne Particles from Different Regions of a Megacity on In Vitro Secretion of Cytokines by a Monocyte Line during Different Seasons

**DOI:** 10.3390/toxics12020149

**Published:** 2024-02-15

**Authors:** Noemi Meraz-Cruz, Natalia Manzano-León, Daniel Eduardo Sandoval-Colin, María del Carmen García de León Méndez, Raúl Quintana-Belmares, Laura Sevilla Tapia, Alvaro R. Osornio-Vargas, Miatta A. Buxton, Marie S. O’Neill, Felipe Vadillo-Ortega

**Affiliations:** 1Unidad de Vinculación Científica de la Facultad de Medicina, UNAM en el Instituto Nacional de Medicina Genómica, Mexico City 14610, Mexico; nmc@unam.mx (N.M.-C.); e.sandoval.colin@gmail.com (D.E.S.-C.); mcglm@hotmail.com (M.d.C.G.d.L.M.); 2Subdirección de Investigación Básica, Instituto Nacional de Cancerología, Mexico City 14080, Mexico; natymleon@hotmail.com (N.M.-L.); qbro76@gmail.com (R.Q.-B.); lauratapia@gmail.com (L.S.T.); 3Department of Pediatrics, Faculty of Medicine and Dentistry, University of Alberta, Edmonton, AB T6G 1C9, Canada; osornio@ualberta.ca; 4Department of Epidemiology, School of Public Health, University of Michigan, Ann Arbor, MI 48109, USA; mabuxton@umich.edu (M.A.B.); marieo@umich.edu (M.S.O.); 5Department of Environmental Sciences, School of Public Health, University of Michigan, Ann Arbor, MI 48109, USA

**Keywords:** air pollution, cytokine, megacity, PM_10_, spatial variation

## Abstract

Several epidemiological studies have demonstrated that particulate matter (PM) in air pollution can be involved in the genesis or aggravation of different cardiovascular, respiratory, perinatal, and cancer diseases. This study assessed the in vitro effects of PM_10_ on the secretion of cytokines by a human monocytic cell line (THP-1). We compared the chemotactic, pro-inflammatory, and anti-inflammatory cytokines induced by PM_10_ collected for two years during three different seasons in five different Mexico City locations. MIP-1α, IP-10, MCP-1, TNF-α, and VEGF were the main secretion products after stimulation with 80 μg/mL of PM_10_ for 24 h. The THP-1 cells showed a differential response to PM_10_ obtained in the different sites of Mexico City. The PM_10_ from the north and the central city areas induced a higher pro-inflammatory cytokine response than those from the south. Seasonal pro-inflammatory cytokine secretion always exceeded anti-inflammatory secretion. The rainy-season-derived particles caused the lowest pro-inflammatory effects. We concluded that toxicological assessment of airborne particles provides evidence supporting their potential role in the chronic exacerbation of local or systemic inflammatory responses that may worsen the evolution of some chronic diseases.

## 1. Introduction

Air pollution is a well-known risk factor for adverse human health effects [1,2]. Exposure to high quantities of airborne particulate matter (PM) is associated with pregnancy complications [3,4,5,6] and increased morbidity and mortality from respiratory [7,8,9] and cardiovascular diseases [10,11,12,13]. PM’s proposed damage mechanisms involve the secretion of pro-inflammatory cytokines and direct cytotoxic and genotoxic effects [14,15,16]. Some of these effects are mediated by reactive oxygen species, as demonstrated by in vivo and in vitro studies [17,18,19]. Target organs and health outcomes have been related to three types of particles of different sizes and compositions. The PM_10_ (aerodynamic diameter (AED) < 10 μm) is mainly deposited in the upper airways. In comparison, PM_2.5_ (AED < 2.5 μm) and PM_0.1_ (AED < 0.1 μm) can reach the intravascular compartment from terminal bronchioles and alveoli [20,21]. All these size fractions of PM can induce or aggravate conditions with an inflammatory background. PM also significantly contributed to excess mortality during the COVID-19 pandemic [22,23].

PM composition depends on the sources, human activities, and geographical and meteorological local characteristics. PM can be anthropogenic derived from vehicle emissions, industry, soil erosion or produced naturally during dust storms, forest fires, or volcanic eruptions. Seasonal variations in air pollution and PM-related health effects have been demonstrated [10,24,25]. Indeed, PM’s toxic and inflammatory potential has been shown to vary due to the chemical diversity of its components based on local conditions and the time of the year [26,27,28]. A relationship has been demonstrated between elevated PM_10_ levels and increased mortality, showing evidence even from the assessment of seasonality together with temperature variability in Mexico City [29].

Mexico City is one of the most densely populated cities globally, with 5967 people per km^2^, and it is also among the most polluted cities in Latin America [30]. Given the multifactorial causes of pollution and that 99.5% of the population is urban, Mexico City residents have been exposed to high pollutant levels for decades, with notably high levels of O_3_, PM_2.5_, and PM_10_. Additionally, geographical conditions, including altitude and being located at a tropical latitude within a valley surrounded by mountains, make the city more prone to high air pollution levels. Studies conducted by the air-monitoring network of the Mexico City government showed that the permissible PM_10_ level of 150 μg/m^3^ was generally exceeded in some areas of the city, mainly in correlation with the location of industrial activity [31,32]. In addition, different studies examining PM in Mexico City have shown a seasonal variation in size and chemical composition [33,34].

Most public health recommendations about air pollutant exposure are based on epidemiological findings correlating air quality monitoring with health outcomes, and few efforts are available to evaluate the direct effect of air pollutants on biological responses and to use these results to identify geographical regions or seasonal timeframes with higher risks for health. Therefore, in this study, the human monocyte cell line THP-1 was evaluated as a biomonitor of inflammatory responses to PM_10_ collected in different areas and seasons of a megacity.

## 2. Materials and Methods

**PM sampling**. PM_10_ were collected in five Mexico City locations from March 2010 to February 2012. The ethics committees of Facultad de Medicina, UNAM (102-2009) and the University of Michigan Institutional Review Board (HUM00023514) approved the protocols for sample collection and analysis. Sampling sites were selected based on their proximity to Mexico City’s air monitoring stations in areas representing either dominant industrial, business, or residential activities. The chosen locations vary in traffic-related pollution, demographics, and urban infrastructure: an industrial region located in the north, a business one situated downtown or central, and residential areas in the south, the east, and the west [35].

High-volume air samplers (TE6070V-2.5, Tisch Environmental, Inc.; Hamilton, OH, USA, airflow rate 1.13 m^3^ min^−1^) [14] equipped with modified nitrocellulose membranes were used to collect PM_10_ for 24 h on Mondays, Wednesdays, and Fridays for 24 months. The PM collection periods corresponded to the three seasons observed at Mexico City’s latitude, namely warm-dry (WD): March–May; rainy (R): June–October; and cold-dry (CD): November–February.

**PM sample preparation**. PM was mechanically recovered from the membranes and pooled according to month and site, resulting in 18 samples from the WD season, 30 samples from the R season, and 24 samples from the CD season. Following measurements of their weight, the PM samples were stored individually in baked glass vials and preserved in the dark, at 4 °C, in desiccators. PM samples were sterilized by autoclaving before use for in vitro exposure experiments [33].

**Cell culture**. To evaluate the cell response to PM, we used THP-1 cells (human monocytic cell line) obtained from the American Type Culture Collection (TIB 202). The cell suspensions were grown in RPMI 1640 media (Sigma Chemical, St. Louis, MO, USA) supplemented with 10% fetal bovine serum, containing penicillin (50 U/mL) and streptomycin (50 mg/mL).

One milliliter per well cell suspensions (550,000 cells/mL) were kept in 24-well plates at 37 °C in a 5% CO_2_/95% air atmosphere. Culture media was replaced by serum-free media and incubated for 24 h before exposure to PM. In previous work, 80 μg/mL was the optimal PM concentration to induce cytokine production with minimal cell viability loss (up to 10%) [33]. PM stock suspensions (1.0 mg/mL) were prepared in cell culture media, sonicated for 5 min, and vortexed before addition to cell cultures to reach a final concentration of 80 μg/mL. THP-1 cells were incubated for 24 h in the presence of PM, then centrifuged at 2000× *g*, and supernatants were recovered and maintained at −80 °C until use for cytokine quantification. Three independent experiments were carried out in triplicate with each PM sample. Non-exposed cells were used as negative controls, and their basal cytokine levels were subtracted from the experimental values. 

**Multiplex for cytokines/chemokines**. A panel of fifteen cytokines (MAP human cytokine/chemokine magnetic bead panel kit; Millipore Corporation, Billerica, MA, USA) was used, including Eotaxin, interleukin (IL), IL-10, IL-17, IL-2, IL-6, IL-12p40, IL-1α, IL-1β, interleukin-1 receptor antagonist (IL-1RA), soluble interleukin-2 receptor alpha (sIL-2Rα), interferon-gamma-induced protein 10 (IP-10), monocyte chemoattractant protein-1 (MCP-1), macrophage inflammatory protein (MIP-1α), tumor necrosis factor-alpha (TNF-α), and vascular endothelial growth factor (VEGF). Multiplex analyses were performed in samples kept at −80 °C for no more than 90 days, following the published protocol of the manufacturer, and concentrations are expressed in pg/mL.

**Statistical Analysis**. Cytokine concentrations in culture media of PM-stimulated THP-1 were averaged according to each triplicate. Descriptive statistics were calculated for each of the city’s five locations and seasons of the year. Natural log (ln)-transformation of cytokine data was performed to approximate normality. The Shapiro–Wilk test and Anderson–Darling test were used to assess continuous data for normality, and the Levene’s and Breusch–Pagan tests were used to evaluate heteroskedasticity. One-way analysis of variance (ANOVA) with post hoc test Tukey’s HSD was used to test differences across regions and seasons. A supervised clustering heatmap was performed separating cytokine families by regions and seasons. Concentrations were standardized, and the Spearman correlation distance measure was applied to cluster analysis. 

Principal component analysis (PCA) was used to characterize a cytokine profile to indicate the level of inflammatory balance in the cellular response. An iterative process was used to reduce cytokines’ dimensionality and lead to the maximum variance between cytokine families and city-regions. To achieve this, data were normalized and centered with a mean of 0 and a standard deviation of 1. The Kaiser–Guttman criterion was used to choose the number of components to retain [36]. The cytokines selected for the final PCA explained the highest variance within two principal components; they defined the best profile that maximized the variance between the city sectors (one chemotactic, two pro-inflammatory, and one anti-inflammatory). Principal component scores were compared between city regions using the Kruskal–Wallis rank-sum test with Dunn’s multiple comparison test. A Mexico City region-level choropleth map indicating distinct areas according to the balance between the selected cytokines was created. To calculate the proinflammatory (IL-1α, IL-β, TNF-α, IL-6) ratio over the anti-inflammatory (IL-1RA, IL-10) in the different regions, simple ratios were calculated in each of them, according to the values reported in Table 1.

## 3. Results

Eotaxin, IL-2, IL-12p40, IL-17, and sIL-2Ra were not included in the analysis because their concentrations in the culture media were below the detection limit. MIP-1α, IP-10, MCP-1, TNF-α, and VEGF were the main secretion products of THP-1 cells upon PM_10_ stimulation. IL-6 had a minor contribution to THP-1 responses. Cytokines were grouped for analysis according to their main biological activity: chemotactic (MIP-1α, IP-10, and MCP-1), pro-inflammatory (IL-1α, IL-1β, TNF-α, and IL-6), and anti-inflammatory (IL-1RA and IL-10). THP-1 cells showed differential responses to PM_10_ obtained from different regions of Mexico City and during the two years of follow-up (Figure 1; Table 1).

Chemokine averaged concentrations were higher with PM_10_ obtained in the north, central, and west areas and lowered with particles from the south (Figure 2; Table 1). In addition, the secretion of MIP-1α was twice as high when stimulated with PM_10_ from the north, central, and west compared with the response induced by particles from the south or east (Figure 2, Table 1 and Table 2). 

PM_10_ induced differential location-based pro-inflammatory cytokine production. Particles from the south and central generated higher values of IL-1β and IL-6. However, IL-1α was mainly caused by particles from the north and west (Figure 2, Table 1 and Table 2), and particles from the central induced the highest levels of TNF-α. 

Particles from the north caused less secretion of the anti-inflammatory cytokines IL-10 and IL-1RA. VEGF did not show differential responses (Figure 2). 

City-averaged values of particle-induced chemokines showed only a significant increase in chemotactic (MCP-1) during the warm-dry season (WD) (Figure 3). Consistently higher values of pro-inflammatory cytokines, except for IL-6, were induced by particles collected during warm-dry and cold-dry seasons (Figure 3, Table 3 and Table 4). No major differences in anti-inflammatory cytokine secretion were found between seasons. VEGF was increased in the media culture of rainy season stimulated cells (Figure 3). 

According to the function of the groups of cytokines previously mentioned, an iterative selection was made using principal component analysis which allowed us to simplify the data set for all cytokines secreted by THP-1 cells after being stimulated by particles from different regions during two years of observation. Representative cytokines with direct biological significance, including chemokine (MIP-1α), proinflammatory (IL-1α and IL-1β), and anti-inflammatory (IL-1RA), resulted from this analysis. Using this combination of cytokines, three clearly differentiated regions were identified: first the northern region, characterized by the highest proinflammatory particles; second the central and west; and third comprising the southern and eastern regions (Figure 4). Principal component scores reflecting the balance between proinflammatory and anti-inflammatory cytokines revealed that the northern zone induced the highest proinflammatory response and the southern and eastern the lowest (Figure 5, Table 5 and Table 6). 

## 4. Discussion

We successfully used THP-1 cells as an experimental model for evaluating the innate immune system’s response to airborne PM_10_. This monocytic human leukemia cell line has been widely used to study the functions, mechanisms, signaling pathways, nutrient, and drug transport of monocytes/macrophages [37,38], and we propose it can be used as a proxy of the inflammatory responses induced by airborne PM_10_ obtained in places such as Mexico City. Previous descriptions of THP-1 confirm that they can recognize, internalize, and process PM, activating NOD-like receptors (NLR) that, in turn, induce the activation of transcription factors, such as Nuclear Factor kB (NF-κB), AP-1, and SP-1 [38], and the signal transducer and activator of transcription-1 (STAT1) [39]. This activation results in the up-regulation of pro-inflammatory factors, such as TNF-α and IL-6, and the protein NLR3, necessary for the assembly of the inflammasome and the proteolytic activation of the pro-inflammatory cytokines IL-1β and IL-18 [40]. All these characteristics make THP-1 a well-characterized cell line that can be used to biomonitor the inflammation-mediated responses to air pollutants and to correlate these responses with health outcomes.

This study allowed the collection of PM_10_ over two years, simultaneously obtaining particles from five different sectors of a large city and evaluating them with the THP-1 biological system. Cytokine secretion patterns by THP-1 cells were consistent with the environmental characteristics of each region of the city. Sites with the highest industrial and pollution levels and a high population density, such as the northern and central sectors [41], induced a predominant pro-inflammatory effect on cells, calculated by the net amount of secreted chemokines and pro-inflammatory cytokines and as the ratio of pro-inflammatory/anti-inflammatory cytokines. Particles captured in areas of the city with well-preserved green spaces and little industrial activity showed a low capacity for the induction of cytokine secretion. Cytokine secretion by stimulated THP-1 cells also varied according to the season when the particles were collected, probably due to changes in their composition, which has been reported in previous studies [25,33,42]. 

Air pollution composition in Mexico City has cycles associated with changes in humidity, temperature, and precipitation levels in each season and other factors such as the intensity of vehicular traffic, industrial activity, construction, and demolition activities, which modify the content of several chemicals and diverse biological-derived materials. In the dry-cold season, pollution levels increase, with PM being higher in winter, while ozone is more elevated in the rainy season [43,44]. Mexico City is located in a basin, and the winds entering from the north and west disperse the particulate components in a gradient from north to south [45]. According to these changes, chemokines and pro-inflammatory cytokines were secreted by THP-1 cells in low concentrations when cells were exposed to particles captured during the rainy season. On the contrary, higher values were observed when using particles from the cold-dry season, supporting the existence of different city microenvironments in which we may expect non-identical exposure to pollutants and dissimilar extent and severity of health effects [46]. 

In this study, no attempt to characterize the composition of particles was made because they have been extensively studied in the past [33,47,48,49]. Studies by Manzano-Leon et al. in 2013 and 2016 [27,33], coupled with Rosas-Pérez et al. [48], show that the chemical properties of particles are consistent and induce proinflammatory responses resulting from complex interactions between PM constituents. In this study, we validate regional differential effects on the secretion of several cytokines, confirming region-specific contributions of particle constituents in THP-1 cells. Previous studies in Mexico City have shown that particles obtained from the northern area, but not those obtained from the south part of the city, induced cell death and DNA damage in proliferating cells and may be related to the high content of metals [50]. Pro-inflammatory effects of PM_10_ became more noticeable in the central zone, which may be the result of a synergy between metals, PAHs, and endotoxins. On the contrary, in the southern zone, the lowest proinflammatory effect of particles was identified, in coincidence with low levels of metals and PAHs in particles collected in this area in previous studies. In addition, PM_10_ levels are always lower in this location, which is characterized by relatively well-preserved extensive green areas. 

The adverse health effects of PM are a consequence of its physical characteristics (size, mass, and even shape) and the chemical components they contain, such as polycyclic aromatic hydrocarbons (PAHs), heavy metals (HM), and biogenic materials such as lipopolysaccharide (LPS), all of them with potential pro-inflammatory effects. The response of THP-1 cells to PAHs and dioxins is mediated by the aryl hydrocarbon receptor (AhR), a highly conserved intracellular transcription factor that interacts with NF-kB [51] through physical association and transcriptional modulation. AhR, when activated in the presence of high concentrations of PAHs, a situation that may be induced by PM_10_ from the northern and central zones in our study, induces increased TNFα secretion by a mechanism related to MAPK and ERK [52,53]. On the other hand, NF-kB activity negatively regulates the inflammatory response mediated by LPS in monocytes and macrophages, through its interaction with Stat1 [39]. Thus, it induces the negative regulation of NLRP3, acting as a physiological suppressor of NLRP3 inflammasome and caspase-1 activation, directly affecting IL-1β secretion [54]. PM_10_ with low PAHs and high LPS concentrations such as those collected in the southern region results in preferential NF-kB activation that inhibits the activation of the AhR pathway [55]. 

A proposed biomonitor, such as THP-1 monocytes that evaluate the pro-inflammatory effects of particles, must be correlated with clinical and epidemiological outcomes. Cytokine secretion by THP-1 in response to PM_10_ is direct evidence that airborne particles can contribute to increased inflammatory factors and cellular recruitment in the lung, which promotes physiology alterations, resulting in enhanced acute respiratory symptoms as chronic obstructive pulmonary disease and asthma, pulmonary and systemic oxidative stress, and inflammation [56,57]. Likewise, PM can activate other cellular mediators that produce pulmonary fibrosis. All components present in the particle form a final complex mixture that will produce or activate inflammatory processes, damage, or oxygen-containing reactive species (ROS) in the lung. All these changes harm the epithelium, increasing epithelial permeability. In addition, once airway macrophages have phagocytized PM_10_, the macrophages can divert some of these cytokines to the systemic circulation, explaining the long-distance effects of the cytokines in the cardiovascular system, modulating carcinogenesis [43,58,59] and inducing perinatal complications such as preterm labor or premature rupture of the membranes [60].

Recent epidemiological evidence of the COVID-19 pandemic in Mexico City suggests that air pollution may explain excess mortality in cities with high pollution levels [22,23]. Considering that inflammation is a primary contributor to morbidity and mortality, conditioned by the innate immune response to SARS-CoV-2, our results provide direct evidence that PM_10_ can induce additional pro-inflammatory responses that may explain the asymmetry in both COVID-19-related excess mortality and morbidity during different seasons of the year [61]. Identifying areas of the city with the highest capacity to induce inflammatory responses by the THP-1 bioindicator may aid in developing focalized government efforts to modify air pollution sources and consequently lower population exposure to air pollutants [62].

## 5. Conclusions

We validated the potential use of THP-1 as a biomonitor for the inflammatory effects of PM_10_. It was interesting to observe that there are differences in the secretion of cytokines according to the geographical region and season from which the particulate material comes. Using data reduction techniques, we also identified four possible ‘sentinel’ cytokines that can indicate the level of inflammatory balance in the cellular response, potentially reducing the analysis costs and allowing for regular monitoring of the biological effects of PM by season and location.

Our results support the hypothesis that particle composition may explain differences in inflammation and toxic responses induced by air pollution in different sectors and seasons in a megacity. This is relevant as it would help to carry out preventive measures in the population according to where they live and the season of the year [63], as well as in the treatment of people with different respiratory diseases exposed to particulate matter.

## Figures and Tables

**Figure 1 toxics-12-00149-f001:**
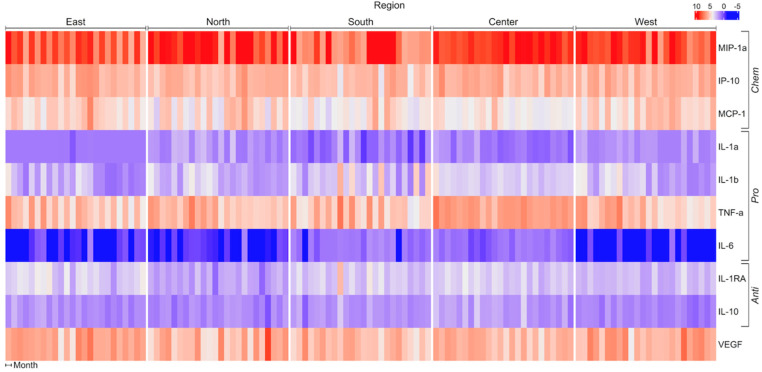
Matrix showing change in pro-, anti-, and chemotactic cytokines when THP−1 cells are incubated with PM_10_ from different regions of Mexico City. Concentration units (pg/mL) were standardized, and the Spearman correlation distance measure was applied to cluster analysis. IL, interleukin; IP10, induced protein-10; MCP, monocyte chemotactic protein; MIP, macrophage inflammatory protein; TNF, tumor necrosis factor; VEGF, vascular endothelial growth factor.

**Figure 2 toxics-12-00149-f002:**
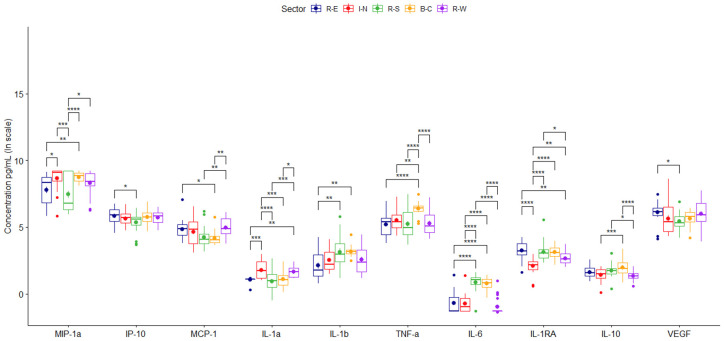
Concentrations of cytokines secreted by TPH-1 cells stimulated by PM_10_. Comparison between the average values obtained for each of the cytokines secreted by samples collected during two years in different megacity regions. Data are presented as medians (middle line) and first and third quartiles (boxes) on a logarithm scale of pg/mL. Units in picograms per milliliter (pg/mL). Significance codes: * (*p* < 0.05), ** (*p* ≤ 0.01), *** (*p* ≤ 0.001) or **** (*p* ≤ 0.0001).

**Figure 3 toxics-12-00149-f003:**
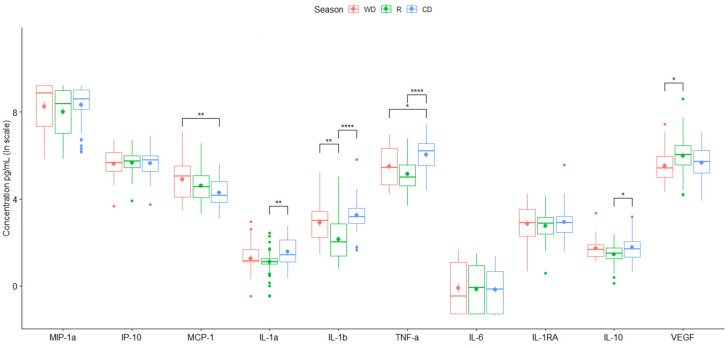
Mean value of cytokine secretion by the THP-1 cell line induced by PM_10_ collected in the different monitoring stations for two years. Data are presented as medians (middle line) and first and third quartiles (boxes) in a natural logarithm scale of pg/mL. One-way analysis of variance (ANOVA) and Tukey’s HSD post hoc analysis were used to test differences across seasons. Significance codes: * (*p* ≤ 0.05), ** (*p* ≤ 0.01) or **** (*p* ≤ 0.0001).

**Figure 4 toxics-12-00149-f004:**
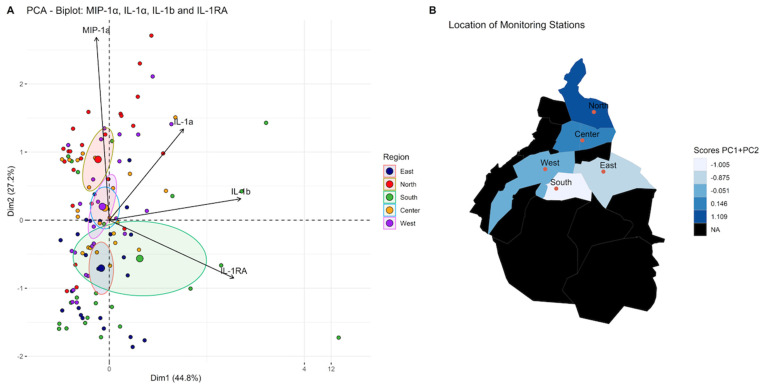
Iterative principal component analysis of cytokines secreted by TPH−1 cells stimulated by PM_10_ from the different regions of the city selected four main contributors for variance: (**A**). Principal component 1(PC1, Dim 1) comprises IL-1b and IL-1RA. Principal component 2 (PC2, Dim 2) comprises MIP-1a and IL-1a. (**B**). Comparison of scores PC1 + PC2 in different regions of the megacity. The distribution pattern obtained is shown. With these cytokines, three clusters are identified: the first included the north, the second was central and west regions, and the last one was in the south and east regions. The black areas on the map are forest zones, almost not urbanized.

**Figure 5 toxics-12-00149-f005:**
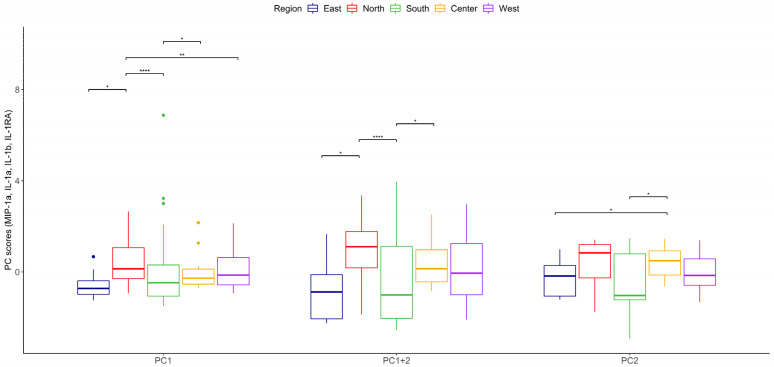
Comparison of PC1 + PC2 scores among regions of the megacity, Kruskal–Wallis test. Statistically significant changes relative to PC1 + PC2 scores among regions are shown with * (*p* ≤ 0.05), ** (*p* ≤ 0.01) or **** (*p* ≤ 0.0001).

**Table 1 toxics-12-00149-t001:** Descriptive statistics for cytokine production over the course of two years in response to PM exposure according to Sectors.

	Sector				
	Residential-East (*n* = 24)	Industrial-North (*n* = 24)	Residential-South (*n* = 24)	Business-Center (*n* = 24)	Residential-West (*n* = 24)
Cytokine	Mean (95% CI ^a^)	Mean (95% CI ^a^)	Mean (95% CI ^a^)	Mean (95% CI ^a^)	Mean (95% CI ^a^)
**Chemotactic**					
MIP-1α	3753.16 (2558.64, 4947.68)	7208.64 (5737.58, 8679.70)	3731.48 (1933.75, 5529.22)	6687.41 (5671.14, 7703.68)	5371.56 (3996.32, 6746.81)
IP-10	401.28 (305.08, 497.48)	314.71 (249.20, 380.22)	251.75 (193.87, 309.63)	360.32 (278.25, 442.39)	341.20 (273.55, 408.85)
MCP-1	170.07 (78.26, 261.89)	155.29 (92.50, 218.08)	98.79 (52.01, 145.57)	74.78 (51.14, 98.43)	175.82 (128.74, 222.89)
**Pro-inflammatory**					
IL-1α	2.93 (2.78, 3.07)	7.11 (5.10, 9.12)	3.57 (2.10, 5.05)	3.57 (2.53, 4.60)	5.62 (4.55, 6.69)
IL-1β	13.96 (7.11, 20.81)	17.65 (10.94, 24.37)	52.41 (19.31, 85.52)	26.38 (20.29, 32.48)	21.14 (11.94, 30.34)
TNF-α	266.62 (160.49, 372.76)	323.55 (197.03, 450.06)	318.79 (152.66, 484.91)	652.74 (525, 780.47)	301.23 (164.04, 438.43)
IL-6	0.71 (0.35, 1.06)	0.64 (0.33, 0.96)	2.79 (2.26, 3.31)	2.37 (1.97, 2.77)	0.49 (0.26, 0.72)
**Anti-inflammatory**					
IL-1RA	30.59 (23.32, 37.86)	9.13 (7.26, 11.01)	33.46 (12.26, 54.67)	24.50 (20.01, 28.98)	15.68 (12.36, 18.99)
IL-10	5.58 (4.48, 6.68)	4.55 (3.81, 5.29)	6.46 (4.82, 8.11)	8.95 (6.17, 11.73)	4.01 (3.40, 4.62)
**Growth factor**					
VEGF	551.93 (405.15, 698.70)	566.49 (110.62, 1022.35)	283.30 (193.44, 373.16)	324.72 (259.72, 389.73)	569.17 (353.70, 784.64)

MIP-1α, macrophage inflammatory protein-1 alpha; IP-10, interferon-γ-inducible protein 10; MCP-1, monocyte chemoattractant protein-1; IL-1α, interleukin-1 alpha; IL-1β, interleukin-1 beta; TNF-α, tumor necrosis factor alpha; IL-6, interleukin-6; IL-1RA, interleukin-1 receptor antagonist; IL-10, interleukin-10; VEGF, vascular endothelial growth factor. Units in picograms per mL (pg/mL). ^a^ 95% confidence interval [$\bar{x}\pm \;\frac{S}{\sqrt{n}}t_{\propto /2}(n-1$)].

**Table 2 toxics-12-00149-t002:** Differences across Sectors over the course of two years according to cytokine production in response to PM exposure.

		Sector									
		R-E, I-N	R-E, R-S	R-E, B-C	R-E, R-W	I-N, R-S	I-N, B-C	I-N, R-W	R-S, B-C	R-S, R-W	B-C, R-W
Chemotactic											
MIP-1α	DBM (95% CI)	0.87 (0.09, 1.64)	−0.32 (−1.10, 0.45)	0.96 (0.19, 1.74)	0.53 ((−0.25,1.30)	−1.19 (−1.96, −0.42)	0.10 (−0.68, 0.87)	−0.34 (−1.11, 0.43)	1.29 (0.51, 2.06)	0.85 (0.08, 1.62)	−0.44 (−1.21, 0.34)
	*p*-value	1.97 × 10^−2^ *	7.75 × 10^−1^	6.76 × 10^−^^3^ **	3.28 × 10^−1^	3.92 × 10^−4^ ***	9.97 × 10^−1^	7.42 × 10^−1^	9.99 × 10^−5^ ****	2.35 × 10^−2^ *	5.22 × 10^−1^
IP-10	DBM (95% CI)	−0.18 (−0.64, 0.29)	−0.48 (−0.95, −0.02)	−0.06 (−0.53, 0.41)	−0.11 (0.57, 0.36)	−0.31 (−0.77, 0.16)	0.12 (−0.35, 0.58)	0.07 (−0.39, 0.54)	0.42 (−0.04, 0.89)	0.38 (−0.09, 0.85)	−0.05 (−0.51, 0.42)
	*p*-value	8.29 × 10^−1^	3.83 × 10^−2^ *	9.97 × 10^−1^	9.71 × 10^−1^	3.68 × 10^−1^	9.56 × 10^−1^	9.93 × 10^−1^	9.37 × 10^−2^	1.70 × 10^−1^	9.99 × 10^−1^
MCP-1	DBM (95% CI)	−0.18 (−0.75, 0.40)	−0.57 (−1.15, 0.01)	−0.66 (−1.23, −0.08)	0.13 (−0.45, 0.71)	−0.39 (−0.97, 0.19)	−0.48 (−1.06, 0.10)	0.31 (−0.27, 0.88)	−0.09 (−0.66, 0.49)	0.70 (0.12, 1.28)	0.79 (0.21, 1.36)
	*p*-value	9.15 × 10^−1^	5.58 × 10^−2^ *	1.75 × 10^−2^ *	9.71 × 10^−1^	3.33 × 10^−1^	1.52 × 10^−1^	5.83 × 10^−1^	9.93 × 10^−1^	9.41 × 10^−^^3^ **	2.36 × 10^−^^3^ **
Pro-inflammatory											
IL-1α	DBM (95% CI)	0.70 (0.24, 1.16)	−0.14 (−0.60, 0.32)	0.03 (−0.43, 0.49)	0.57 (0.12, 1.03)	−0.84 (−1.30, −0.38)	−0.67 (−1.13, −0.21)	−0.13 (−0.59, 0.33)	0.17 (−0.29, 0.63)	0.71 (0.25, 1.17)	0.54 (0.08, 1.00)
	*p*-value	4.30 × 10^−4^ ***	9.18 × 10^−1^	1.00	6.44 × 10^−^^3^ **	1.44 × 10^−5^ ****	9.02 × 10^−4^ ***	9.39 × 10^−1^	8.37 × 10^−1^	3.28 × 10^−4^ ***	1.20 × 10^−2^ *
IL-1β	DBM (95% CI)	0.42 (−0.34, 1.19)	1.02 (0.26, 1.78)	1.07 (0.30, 1.83)	0.44 (−0.32, 1.20)	0.60 (−0.16, 1.36)	0.64 (−0.12, 1.41)	0.02 (−0.74, 0.78)	0.04 (−0.72, 0.81)	−0.58 (−1.34, 0.18)	−0.62 (−1.39, 0.14)
	*p*-value	5.45 × 10^−1^	2.94 × 10^−^^3^ **	1.69 × 10^−^^3^ **	4.99 × 10^−1^	1.96 × 10^−1^	1.41 × 10^−1^	1.00	1.00	2.25 × 10^−1^	1.64 × 10^−1^
TNF-α	DBM (95% CI)	0.29 (−0.35, 0.94)	0.02 (−0.62, 0.67)	1.18 (0.53, 1.83)	0.07 (−0.58, 0.71)	−0.27 (−0.92, 0.38)	0.88 (0.24, 1.53)	−0.23 (−0.87, 0.42)	1.16 (0.51, 1.80)	0.04 (−0.60, 0.69)	−1.11 (−1.76, −0.46)
	*p*-value	7.18 × 10^−1^	1.00	1.7 × 10^−5^ ****	9.99 × 10^−1^	7.75 × 10^−1^	2.25 × 10^−3^ **	8.68 × 10^−1^	2.57 × 10^−5^ ****	1.00	5.62 × 10^−5^ ****
IL-6	DBM (95% CI)	−0.02 (−0.55, 0.51)	1.55 (1.02, 2.09)	1.48 (0.95, 2.01)	−0.28 (−0.81, 0.25)	1.58 (1.04, 2.11)	1.50 (0.97, 2.03)	−0.26 (−0.79, 0.27)	−0.08 (−0.61, 0.45)	−1.84 (−2.37, −1.31)	−1.76 (−2.29, −1.23)
	*p*-value	1.00	5.88 × 10^−12^ ****	4.7 × 10^−11^ ****	5.80 × 10^−1^	3.34 × 10^−12^ ****	2.6 × 10^−11^ ****	6.49 × 10^−1^	9.95 × 10^−1^	4.79 × 10^−14^ ****	6.59 × 10^−14^ ****
Anti-inflammatory											
IL-1RA	DBM (95% CI)	−1.17 (−1.63, −0.71)	−0.11 (−0.57, 0.35)	−0.13 (−0.59, 0.32)	−0.59 (−1.05, −0.13)	1.06 (0.60, 1.52)	1.03 (0.57, 1.49)	0.58 (0.12, 1.04)	−0.03 (−0.49, 0.43)	−0.48 (−0.94, −0.02)	−0.46 (−0.92, 0.00)
	*p*-value	1.46 × 10^−9^ ****	9.66 × 10^−1^	9.26 × 10^−1^	4.73 × 10^−^^3^ **	3.57 × 10^−8^ ****	7.8 × 10^−8^ ****	6.30 × 10^−^^3^ **	1.00	3.39 × 10^−2^ *	5.26 × 10^−2^ *
IL-10	DBM (95% CI)	−0.20 (−0.58, 0.18)	0.11 (−0.27, 0.50)	0.36 (−0.02, 0.75)	−0.30 (−0.68, 0.08)	0.31 (−0.07, 0.70)	0.56 (0.18, 0.95)	−0.10 (−0.48, 0.28)	0.25 (−0.14, 0.63)	−0.41 (−0.80, −0.03)	−0.66 (−1.05, −0.28)
	*p*-value	5.97 × 10^−1^	9.25 × 10^−1^	7.52 × 10^−2^ *	2.00 × 10^−1^	1.64 × 10^−1^	8.45 × 10^−4^ ***	9.52 × 10^−1^	3.83 × 10^−1^	2.82 × 10^−2^ *	5.17 × 10^−5^ ****
Growth factor											
VEGF	DBM (95% CI)	−0.48 (−1.12, 0.17)	−0.69 (−1.34, −0.05)	−0.47 (−1.12, 0.17)	−0.11 (−0.76, 0.53)	−0.21 (−0.86, 0.43)	0.00 (−0.64, 0.65)	0.37 (−0.28, 1.01)	0.22 (−0.43, 0.86)	0.58 (−0.06, 1.23)	0.36 (−0.28, 1.01)
	*p*-value	2.47 × 10^−1^	2.89 × 10^−2^ *	2.56 × 10^−1^	9.89 × 10^−1^	8.88 × 10^−1^	1.00	5.18 × 10^−1^	8.81 × 10^−1^	9.93 × 10^−2^ *	5.30 × 10^−1^

R-E, Residential-East; I-N, Industrial-North; R-S, Residential-South; B-C, Business-Center; R-W, Residential-West; CI, confidence interval; DBM, difference between means; MIP-1α, macrophage inflammatory protein-1 alpha; IP-10, interferon-γ-inducible protein 10; MCP-1, monocyte chemoattractant protein-1; IL-1α, interleukin-1 alpha; IL-1β, interleukin-1 beta; TNF-α, tumor necrosis factor alpha; IL-6, interleukin-6; IL-1RA, interleukin-1 receptor antagonist; IL-10, interleukin-10; VEGF, vascular endothelial growth factor. Comparisons of means between Sectors were performed using one-way analysis of variance (ANOVA) and Tukey’s HSD test as post hoc analysis on natural log (ln)-transformed cytokines. Statistically significant changes relative to Sectors within each cytokine are shown with * (*p* ≤ 0.05), ** (*p* ≤ 0.01), *** (*p* ≤ 0.001) or **** (*p* ≤ 0.0001).

**Table 3 toxics-12-00149-t003:** Descriptive statistics for cytokine production over the course of two years in response to PM exposure according to Seasons.

	Season		
	Warm-Dry (*n* = 30)	Rainy (*n* = 50)	Cold-Dry (*n* = 40)
Cytokine	Mean (95% CI ^a^)	Mean (95% CI ^a^)	Mean (95% CI ^a^)
Chemotactic			
MIP-1α	5918.32 (4339.06, 7497.59)	4817.26 (3320.05, 6314.48)	5591.04 (4231.61, 6950.47)
IP-10	337.36 (250.81, 423.91)	329.24 (265.28, 393.19)	336.99 (255.69, 418.30)
MCP-1	197.98 (109.46, 286.51)	135.27 (83.42, 187.13)	87.27 (63.10, 111.44)
Pro-inflammatory			
IL-1α	4.55 (2.89, 6.21)	3.51 (2.68, 4.33)	5.88 (4.26, 7.51)
IL-1β	28.93 (14.02, 43.83)	15.65 (4.03, 27.26)	37.67 (16.24, 59.10)
TNF-α	348.83 (230.86, 466.81)	236.71 (148.80, 324.63)	560.24 (383.38, 737.10)
IL-6	1.60 (0.93, 2.27)	1.40 (0.87, 1.93)	1.25 (0.79, 1.72)
Anti-inflammatory			
IL-1RA	22.80 (16.19, 29.41)	18.79 (14.09, 23.49)	27.43 (10.66, 44.20)
IL-10	6.35 (4.39, 8.31)	4.70 (3.86, 5.54)	7.09 (4.99, 9.19)
Growth factor			
VEGF	358.02 (203.39, 512.65)	590.60 (261.16, 920.03)	370.60 (263.45, 477.76)

MIP-1α, macrophage inflammatory protein-1 alpha; IP-10, interferon-γ-inducible protein 10; MCP-1, monocyte chemoattractant protein-1; IL-1α, interleukin-1 alpha; IL-1β, interleukin-1 beta; TNF-α, tumor necrosis factor alpha; IL-6, interleukin-6; IL-1RA, interleukin-1 receptor antagonist; IL-10, interleukin-10; VEGF, vascular endothelial growth factor. Units in picograms per milliliter (pg/mL). ^a^ 95% confidence interval [$\bar{x}\pm \;\frac{S}{\sqrt{n}}t_{\propto /2}(n-1$)].

**Table 4 toxics-12-00149-t004:** Differences across Seasons over the course of two years according to cytokine production in response to PM exposure.

		Season		
		WD, R	WD, CD	R, CD
**Chemotactic**				
MIP-1α	DBM (95% CI)	−0.25 (−0.84, 0.34)	0.06 (−0.56, 0.67)	0.31 (−0.23, 0.85)
	*p*-value	5.78 × 10^−1^	9.72 × 10^−1^	3.71 × 10^−1^
IP-10	DBM (95% CI)	0.06 (−0.27, 0.39)	0.03 (−0.32, 0.38)	−0.03 (−0.33, 0.27)
	*p*-value	9.03 × 10^−1^	9.76 × 10^−1^	9.71 × 10^−1^
MCP-1	DBM (95% CI)	−0.29 (−0.69, 0.12)	−0.61 (−1.04, −0.19)	−0.33 (−0.70, 0.05)
	*p*-value	2.21 × 10^−1^	2.46 × 10^−3^ **	9.91 × 10^−2^ *
Pro-inflammatory				
IL-1α	DBM (95% CI)	−0.15 (−0.50, 0.20)	0.30 (−0.06, 0.67)	0.45 (0.13, 0.77)
	*p*-value	5.68 × 10^−1^	1.18 × 10^−1^	2.83 × 10^−3^ **
IL-1β	DBM (95% CI)	−0.76 (−1.26, −0.27)	0.34 (−0.17, 0.86)	1.11 (0.65, 1.56)
	*p*-value	1.12 × 10^−3^ **	2.61 × 10^−1^	1.98 × 10^−7^ ****
TNF-α	DBM (95% CI)	−0.36 (−0.82, 0.09)	0.52 (0.05, 1.00)	0.89 (0.47, 1.31)
	*p*-value	1.46 × 10^−1^	2.82 × 10^−2^ *	5.62 × 10^−6^ ****
IL-6	DBM (95% CI)	−0.04 (−0.62, 0.53)	−0.07 (−0.67, 0.53)	−0.03 (−0.55, 0.50)
	*p*-value	9.82 × 10^−1^	9.59 × 10^−1^	9.93 × 10^−1^
Anti-inflammatory				
IL-1RA	DBM (95% CI)	−0.11 (−0.50, 0.28)	0.07 (−0.34, 0.48)	0.19 (−0.17, 0.55)
	*p*-value	7.79 × 10^−1^	9.04 × 10^−1^	4.42 × 10^−1^
IL-10	DBM (95% CI)	−0.26 (−0.54, 0.02)	0.06 (−0.23, 0.35)	0.32 (0.06, 0.58)
	*p*-value	7.67 × 10^−2^ *	8.79 × 10^−1^	1.11 × 10^−2^ *
Growth factor				
VEGF	DBM (95% CI)	0.46 (0.02, 0.91)	0.14 (−0.33, 0.61)	−0.32 (−0.73, 0.09)
	*p*-value	4.08 × 10^−2^ *	7.52 × 10^−1^	1.57 × 10^−1^

WD, warm-dry; R, rainy; CD, cold-dry; CI, confidence interval; DBM, difference between means; MIP-1α, macrophage inflammatory protein-1 alpha; IP-10, interferon-γ-inducible protein 10; MCP-1, monocyte chemoattractant protein-1; IL-1α, interleukin-1 alpha; IL-1β, interleukin-1 beta; TNF-α, tumor necrosis factor alpha; IL-6, interleukin-6; IL-1RA, interleukin-1 receptor antagonist; IL-10, interleukin-10; VEGF, vascular endothelial growth factor. Comparisons of means between Seasons were performed using one-way analysis of variance (ANOVA) and Tukey’s HSD test as post hoc analysis on natural log (ln)-transformed cytokines. Statistically significant changes relative to Seasons within each cytokine are shown with * (*p* ≤ 0.05), ** (*p* ≤ 0.01) or **** (*p* ≤ 0.0001).

**Table 5 toxics-12-00149-t005:** Eigenvalues for principal components selection extracted by principal component analysis of MIP-1α, IL-1 α, IL-1β, and IL-1RA.

	Eigenvalue	Percentage of Variance	Cumulative Percentage of Variance
MIP-1α	1.7923532	44.808831	44.80883
IL-1α	1.0874345	27.185863	71.99469
IL-1β	0.8309943	20.774857	92.76955
IL-1RA	0.2892180	7.230449	100.00000

MIP-1α, macrophage inflammatory protein-1 alpha; IL-1α, interleukin-1 alpha; IL-1β, interleukin-1 beta; IL-1RA, interleukin-1 receptor antagonist. Kaiser–Guttman criterion was used to choose the number of components to retain (eigenvalue > 1).

**Table 6 toxics-12-00149-t006:** Loadings of variables in each of the four principal components extracted by principal component analysis.

	Dim.1	Dim.2	Dim.3	Dim.4
IL-1a	0.50658333	0.4456352	−0.7324773	0.09088234
IL-1b	0.89789048	0.1045237	0.2021667	−0.37682372
MIP-1a	−0.08845469	0.8931446	0.4266453	0.11154469
IL-1RA	0.84952635	−0.2832159	0.2675326	0.35569656

MIP-1α, macrophage inflammatory protein-1 alpha; IL-1α, interleukin-1 alpha; IL-1β, interleukin-1 beta; IL-1RA, interleukin-1 receptor antagonist.

## Data Availability

All relevant data are within the manuscript.

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
