# Peer review of "Effects of PM10 Airborne Particles from Different Regions of a Megacity on In Vitro Secretion of Cytokines by a Monocyte Line during Different Seasons"

_toxics, 2024, doi:10.3390/toxics12020149_

Round 1

Reviewer 1 Report

Comments and Suggestions for Authors

Dear Editor, 

I had a great pleassure to be able to read an article entitled: "Effects of PM10 airborne particles from different regions and 2 seasons of a megacity on in-vitro secretion of cytokines by a 3 monocyte line", in which the authors assessed the in vitro effects of PM <10 μm in aerodynamic diameter (PM10) on the secretion of cytokines by a human monocytic cell line (THP-1). The authors compared the chemotactic, pro-inflammatory, and anti-inflammatory cytokines induced by PM10 collected for two years during three different seasons in five different Mexico City locations. The outhors observed thata the PM10 from the north and the central city areas induced a higher pro-inflammatory cytokine response than those from the south. Also, seasonal pro-inflammatory cytokine secretion always exceeded the anti-inflammatory one. The rainy season-derived particles caused the lowest pro-inflammatory effects. The authors concluded that toxicological assessment of airborne particles provides evidence supporting their potential role in chronic exacerbation of local or systemic inflammatory responses that may worsen the evolution of some chronic diseases.

Main main remarks concerning that text are: 

1) Consider please citing this: The relationship between increased air pollution expressed as PM10 concentration and the frequency of percutaneous coronary interventions in patients with acute coronary syndromes-a seasonal differences. Januszek R, Staszczak B, Siudak Z, BartuÅ› J, Plens K, BartuÅ› S, Dudek D. Environ Sci Pollut Res Int. 2020 Jun;27(17):21320-21330.

2) I do not understand why this is placed in the Materials and Methods section? In my opinion this should be removed and replaced by statments related to the current study: 

"The Materials and Methods should be described with sufficient details to allow others to replicate and build on the published results. Please note that the publication of your manuscript implicates that you must make all materials, data, computer code, and protocols associated with the publication available to readers. Please disclose at the submission stage any restrictions on the availability of materials or information. New methods and protocols should be described in detail while well-established methods can be briefly described and appropriately cited. Research manuscripts reporting large datasets that are deposited in a publicly available database should specify where the data have been deposited and provide the relevant accession numbers. If the accession numbers have not yet been obtained at the time of submission, please state that they will be provided during review. They must be provided prior to publication. Interventionary studies involving animals or humans, and other studies that require ethical approval, must list the authority that provided approval and the corresponding ethical approval code."

3) Did the authors take into account the use of measurements by local weather stations?

4) In general this is well written article, and I have no major issues concerning the present study. 

5) In this study, the authors should have the consent number of the bioethics committee and its name.

Author Response

Thank you very much for taking the time to review this manuscript.

Responses to minor revisions:

“comments”

Reviewer 1

1) Consider please citing this: The relationship between increased air pollution expressed as PM10 concentration and the frequency of percutaneous coronary interventions in patients with acute coronary syndromes-a seasonal differences. Januszek R, Staszczak B, Siudak Z, BartuÅ› J, Plens K, BartuÅ› S, Dudek D. Environ Sci Pollut Res Int. 2020 Jun;27(17):21320-21330.

The reference has been added in the manuscript with the no. 46.

2) I do not understand why this is placed in the Materials and Methods section? In my opinion this should be removed and replaced by statments related to the current study: 

"The Materials and Methods should be described with sufficient details to allow others to replicate and build on the published results. Please note that the publication of your manuscript implicates that you must make all materials, data, computer code, and protocols associated with the publication available to readers. Please disclose at the submission stage any restrictions on the availability of materials or information. New methods and protocols should be described in detail while well-established methods can be briefly described and appropriately cited. Research manuscripts reporting large datasets that are deposited in a publicly available database should specify where the data have been deposited and provide the relevant accession numbers. If the accession numbers have not yet been obtained at the time of submission, please state that they will be provided during review. They must be provided prior to publication. Interventionary studies involving animals or humans, and other studies that require ethical approval, must list the authority that provided approval and the corresponding ethical approval code."

Corrected. It has been removed.

3) Did the authors take into account the use of measurements by local weather stations?

Yes we did.

4) In general this is well written article, and I have no major issues concerning the present study. 

5) In this study, the authors should have the consent number of the bioethics committee and its name.

Added in Materials and Methods: 

"The ethics committees of Facultad de Medicina, UNAM (102-2009) and the University of Michigan Institutional Review Board (HUM00023514) approved the protocols for sample collection and analysis."

Reviewer 2 Report

Comments and Suggestions for Authors

The authors of the manuscript entitled “Effects of PM10 airborne particles from different regions and seasons of a megacity on in-vitro secretion of cytokines by a monocyte line” assessed the in vitro effects of PM10 on the secretion of cytokines by a human monocytic cell line (THP-1), with a two-year dataset during three different seasons in five different Mexico City locations.

The manuscript is scientifically sounding, interesting, and useful not only for the scientific community, but also to policy makers, although in my opinion it has some room for improvement (only minor revisions).

Minor revisions:

-        In the Abstract section, lines 16-17, the authors wrote “[…] the in vitro effects of PM <10 μm in aerodynamic diameter (PM10) […]”, so the sentence has no sense. Please rewrite this sentence.

-        Line 27: I think that a black space must be removed before the word “some”.

-        Line 39: A μ is missing when the authors are explaining the meaning of PM10.

-        Line 53: Reference 2 is not appropriate references to support the correlation of PM10 concentrations with increased mortality in Mediterranean, as that study was performed in Delhi (India). Reference 30 is also not valid to justify that relation of PM10 and increased mortality in Mediterranean, as that study don’t present monthly or seasonal results, so it does not conclude that. Moreover, reference 29 is also not valid, because that paper analyzes the intensity and frequency of dust storms in the Eastern Mediterranean, but don’t study its health effects and don’t mention the statement the authors are making in lines 50-52. Please provide correct references or remove the sentence from line 50 to 52.

-        Lines 73 to 86: All this text must be removed, because it consists on the instructions of the journal to write the Materials and Methods section.

-        Line 89: The authors state that the PM10 collection methodology was described previously in reference 34. That reference leads to a paper by the same authors that this manuscript I am reviewing. In that paper they have samples collected weekly that were pooled by month, site, and size. From lines 97 to 99 the authors described the PM10 sampling methodology, and they said that the collection periods corresponded to three seasons. In my opinion, the methodology is well described in lines 97 to 99 and lines 101 to 106, with some differences from the methodology in reference 34 and, moreover, the reference 34 is a self-citation. Therefore, I suggest rewriting the first sentence of this subsection, removing the reference 34. The use of this reference in line 136 is OK.

-        Lines 153 to 155: There is something wrong in the beginning of this sentence. I think it could be de verb tense. Anyway, the first part of the sentence is difficult to understand, so please rewrite it.

-         Figure 1: In the figure caption, the authors are using PM10, but in the main text they are using PM10. Please write it always in the same format. Please check it throughout the entire manuscript.

-        Line 181: I think the authors must write “Tables 1 and 2” instead of “Tables 1”. Please check it.

-        Line 190: please write a p letter instead of a P to indicate the p-value to inform about the statistical significance, as in line 200.

-        Line 203: Please write “Tables 1 and 2” instead of “Table 1 and 2”.

-        Lines 2010: Please write “Tables 3 and 4” instead of “Table 3 and 4”.

-        Line 251: The authors refer to Figure 6, but there is no Figure 6 in this manuscript. Please check this and rewrite it.

-        Lines 273 to 276: Please remove that paragraph, as it is the instructions on the manuscript template to write de Discussion section.

 -        Normally, the conclusions are not part of the Discussion section, but an independent section. Furthermore, the conclusions should be more elaborate than a single paragraph. Please consider including an independent Conclusions section.

Author Response

Thank you very much for taking the time to review this manuscript.

Responses to minor revisions:

“comments”

Reviewer 2

  1. - In the Abstract section, lines 16-17, the authors wrote “[…] the in vitro effects of PM <10 μm in aerodynamic diameter (PM10) […]”, so the sentence has no sense. Please rewrite this sentence.

This study assessed the in vitro effects of PM10 on the secretion of cytokines by a human monocytic cell line (THP-1).

  1. - Line 27: I think that a black space must be removed before the word “some”.

Space was removed

  1. - Line 39: A μ is missing when the authors are explaining the meaning of PM10.

μ was placed

  1. - Line 53: Reference 2 is not appropriate references to support the correlation of PM10 concentrations with increased mortality in Mediterranean, as that study was performed in Delhi (India). Reference 30 is also not valid to justify that relation of PM10 and increased mortality in Mediterranean, as that study don’t present monthly or seasonal results, so it does not conclude that. Moreover, reference 29 is also not valid, because that paper analyzes the intensity and frequency of dust storms in the Eastern Mediterranean, but don’t study its health effects and don’t mention the statement the authors are making in lines 50-52. Please provide correct references or remove the sentence from line 50 to 52.

Requested references were removed and the correct reference was provided.

  1. - Lines 73 to 86: All this text must be removed, because it consists on the instructions of the journal to write the Materials and Methods section.

Text was removed

  1. - Line 89: The authors state that the PM10 collection methodology was described previously in reference 34. That reference leads to a paper by the same authors that this manuscript I am reviewing. In that paper they have samples collected weekly that were pooled by month, site, and size. From lines 97 to 99 the authors described the PM10 sampling methodology, and they said that the collection periods corresponded to three seasons. In my opinion, the methodology is well described in lines 97 to 99 and lines 101 to 106, with some differences from the methodology in reference 34 and, moreover, the reference 34 is a self-citation. Therefore, I suggest rewriting the first sentence of this subsection, removing the reference 34. The use of this reference in line 136 is OK.

The sentence was written and the reference number was placed where appropriate. Note: As the manuscript was corrected, the reference No. 34 changed to No. 33.

  1. - Lines 153 to 155: There is something wrong in the beginning of this sentence. I think it could be de verb tense. Anyway, the first part of the sentence is difficult to understand, so please rewrite it.

The sentence was rewritten:

“A Mexico City region-level choropleth map indicating distinct areas according to the balance between the selected cytokines was created.”

  1. - Figure 1: In the figure caption, the authors are using PM10, but in the main text they are using PM10. Please write it always in the same format. Please check it throughout the entire manuscript.

PM10 is always in the same format

  1. - Line 181: I think the authors must write “Tables 1 and 2” instead of “Tables 1”. Please check it.

Corrected

  1. - Line 190: please write a p letter instead of a P to indicate the p-value to inform about the statistical significance, as in line 200.

Corrected

  1. - Line 203: Please write “Tables 1 and 2” instead of “Table 1 and 2”.

Corrected

  1. - Lines 2010: Please write “Tables 3 and 4” instead of “Table 3 and 4”.

Corrected

  1. - Line 251: The authors refer to Figure 6, but there is no Figure 6 in this manuscript. Please check this and rewrite it.

Corrected, figure 6 do not exist.

  1. - Lines 273 to 276: Please remove that paragraph, as it is the instructions on the manuscript template to write de Discussion section.

Paragraph was removed

  1. - Normally, the conclusions are not part of the Discussion section, but an independent section. Furthermore, the conclusions should be more elaborate than a single paragraph. Please consider including an independent Conclusions section.

Conclusions section included

Reviewer 3 Report

Comments and Suggestions for Authors

This manuscript assessed the in vitro effects of PM <10 μm in aerodynamic diameter (PM10) on the secretion of cytokines by a human monocytic cell line (THP-1). It's an interesting piece of research. This study provides evidence for toxicological assessment of airborne particles. It shows that particulate matter can cause inflammation in the local or systemic home, and even exacerbate the evolution of some chronic diseases. I recommend it to be published in Toxics after major revisions.

1. Further improve the expression and representativeness of key words.

2. It is suggested to supplement the analysis of the collected PM10 components, or the relevant research results of the physical and chemical characteristics of such particles in the corresponding area.

3. Analysis of the reasons for the different reactions of particles in different regions.

Author Response

Thank you very much for taking the time to review this manuscript.

Response to comments:

  1. Further improve the expression and representativeness of key words.

Keyword representativeness: Air pollution; Cytokine; Megacity; PM10; Spatial variation

  1. It is suggested to supplement the analysis of the collected PM10 components, or the relevant research results of the physical and chemical characteristics of such particles in the corresponding area.

Corrected. References: 27, 33, 47, 48, 49 and 50, correspond to results of relevant investigations of the physical and chemical characteristics of said particles in the corresponding area.

  1. Analysis of the reasons for the different reactions of particles in different regions.

Included in the discussion section in the next lines:

Page 15 of the manuscript:

"Response of THP-1 cells to PAHs and dioxins is mediated by the aryl hydrocarbon receptor (AhR), a highly conserved intracellular transcription factor that interacts with NF-kB [51] through physical association and transcriptional modulation. AhR when activated in the presence of high concentrations of PAHs, a situation that may be induced by PM10 from the northern and central zones in our study, induces increased TNFα secretion by a mechanism related to MAPK and ERK [52,53]. On the other hand, NF-kB activity negatively regulates the inflammatory response mediated by LPS in monocytes and macrophages, through its interaction with Stat1 [39]. Thus, it induces the negative regulation of NLRP3, acting as a physiological suppressor of NLRP3 inflammasome and caspase-1 activation, directly affecting IL-1β secretion [54]. PM10 with low PAHs and high LPS concentrations such as those collected in the southern region results in preferential NF-kB activation that inhibits activation of the AhR pathway [55]."

Round 2

Reviewer 3 Report

Comments and Suggestions for Authors

No further comment.

Comments on the Quality of English Language

fine

Author Response

Dear Dr.

On January 4, responses to the suggested comments were sent.
We received an email again, requesting a response in three days. We are awaiting any additional comments that you may request.

Once again, we appreciate the time dedicated to reviewing this manuscript.